# Incidence of shoulder dislocations in the UK, 1995–2015: a population-based cohort study

Anjali Shah,[1] Andrew Judge,[1,2,3] Antonella Delmestri,[1] Katherine Edwards,[1] Nigel K Arden,[1,2] Daniel Prieto-Alhambra,[1] Tim A Holt,[4] Rafael A Pinedo-Villanueva,[1] Sally Hopewell,[5] Sarah E Lamb,[1] Amar Rangan,[1,6,7] Andrew J Carr,[1] Gary S Collins,[1] Jonathan L Rees[1]

[1]Nuffield Department of Orthopaedics, Rheumatology and Musculoskeletal Sciences, University of Oxford, Oxford, UK
[2]Faculty of Medicine, University of Southampton, Southampton, UK
[3]Musculoskeletal Research Unit, University of Bristol, Bristol, UK
[4]Nuffield Department of Primary Care Health Sciences, University of Oxford, Oxford, UK
[5]Oxford Clinical Trials Research Unit, University of Oxford, Oxford, UK
[6]Orthopaedic Surgery, The James Cook University Hospital, South Tees Hospitals NHS Foundation Trust, Middlesbrough, UK
[7]Department of Health Sciences, University of York, York, UK

**Correspondence to**
Dr Anjali Shah;
anjali.shah@ndorms.ox.ac.uk

## ABSTRACT

**Objective** This cohort study evaluates the unknown age-specific and gender-specific incidence of primary shoulder dislocations in the UK.

**Setting** UK primary care data from the Clinical Practice Research Datalink (CPRD) were used to identify patients aged 16–70 years with a shoulder dislocation during 1995–2015. Coding of primary shoulder dislocations was validated using the CPRD general practitioner questionnaire service.

**Participants** A cohort of 16 763 patients with shoulder dislocation aged 16–70 years during 1995–2015 were identified.

**Primary outcome measure** Incidence rates per 100 000 person-years and 95% CIs were calculated.

**Results** Correct coding of shoulder dislocation within CPRD was 89% (95% CI 83% to 95%), and confirmation that the dislocation was a 'primary' was 76% (95% CI 67% to 85%). Seventy-two percent of shoulder dislocations occurred in men. The overall incidence rate in men was 40.4 per 100 000 person-years (95% CI 40.4 to 40.4), and in women was 15.5 per 100 000 person-years (95% CI 15.5 to 15.5). The highest incidence was observed in men aged 16–20 years (80.5 per 100 000 person-years; 95% CI 80.5 to 80.6). Incidence in women increased with age to a peak of 28.6 per 100 000 person-years among those aged 61–70 years.

**Conclusions** This is the first time the incidence of shoulder dislocations has been studied using primary care data from a national database, and the first time the results for the UK have been produced. While most primary dislocations occurred in young men, an unexpected finding was that the incidence increased in women aged over 50 years, but not in men. The reasons for this are unknown. Further work is commissioned by the National Institute for Health Research to examine treatments and predictors for recurrent shoulder dislocation.

**Study registration** The design of this study was approved by the Independent Scientific Advisory Committee (15_260) for the Medicines & Healthcare products Regulatory Agency.

## INTRODUCTION

Shoulder joint dislocations are the most common joint dislocations seen in hospital

### Strengths and limitations of this study

► This is the first time the incidence of shoulder dislocations has been studied in the UK using a large population-based database from primary care.
► A new finding is that shoulder dislocations in women over the age of 50 years increased to a peak of 28.6 per 100 000 person-years in those aged 61–70 years.
► The coding of shoulder dislocation diagnoses was validated by comparison with medical records, using a general practitioner (GP) questionnaire.
► The validation exercise did only include the 16–35 years age group; however, there is no reason why coding of shoulder dislocations would be different for patients aged up to 70 years.
► While not all GP practices responded to the questionnaire, those that did and did not respond were similar by deprivation, geography and other demographic characteristics.

accident and emergency departments and trauma clinics[1,2] with 80%–97% of traumatic glenohumeral dislocations being anterior.[1–6] Traumatic anterior shoulder dislocation most commonly occurs after traumatic injury in young people and usually results in structural problems such as Bankart lesions[7] and the Hills-Sachs lesion.[5] The joint can remain 'unstable', and redislocation rates have been reported as being as high as 85%[8] or 92%[9] in young people. Surgery is becoming more common as a treatment, especially in sporting athletes, with many surgeons and patients opting for surgery after only one dislocation.

A Swedish population-based study was conducted by Hovelius.[10] This study while small is well cited in the literature and observed that 1.7% of the population aged 18–70 years previously had a shoulder dislocation. In a 25-year follow-up study of patients aged 12–40 years, recurrent dislocation was more common in younger people with 72%

of patients aged 12–22 years suffering another dislocation, which dropped to 27% in those aged 30–40 years.[11] Early studies reported high incidence of shoulder dislocation in military and athletic populations, with young men being at greatest risk.[6 12–15] In Edinburgh, a study of 252 patients aged 15–35 years suffering a shoulder dislocation identified that the most common cause (86%) was playing contact sports.[16] Of these, 60% suffered a repeat dislocation, and the mean time for recurrent dislocation to occur was 13.3 months.

A number of studies report incidences ranging from 11.2 to 26.2 per 100 000 person-years for shoulder dislocations.[2 17–21] In 2010, Zacchilli and Owens[19] examined the incidence of traumatic shoulder dislocation in patients of all ages presenting to a random sample of 100 hospital emergency departments across the USA during 2002–2006, as recorded in the National Electronic Injury Surveillance System. Seventy-two per cent of dislocations had occurred in men, and the highest incidence was observed in 20–29 years (47.8 per 100 000 person-years; 95% CI 41.0 to 54.5). The overall incidence rate in men was 34.9 per 100 000 person-years (95% CI 30.1 to 39.7), and in women was 13.3 per 100 000 person-years (95% CI 11.6 to 15.0).

In 2014, Leroux *et al*[22] evaluated the incidence of primary anterior shoulder dislocation in patients aged 16–70 years who underwent a closed reduction of the shoulder during April 2002–September 2010 in Ontario, Canada. Seventy-four per cent of all shoulder dislocations were in men, and the highest incidence was observed in men aged 16–20 years (98.3 per 100 000 person-years). The overall adjusted incidence in men and women were similar to figures reported by Zacchilli and Owens.[19]

The incidence of primary anterior shoulder dislocations has not been examined using all patients in a national dataset. The incidence of shoulder dislocations in the UK remains unknown, and a large population-based UK study has not been previously undertaken.

The aim of this study was to evaluate the age-specific and gender-specific incidence of primary shoulder dislocations as recorded in primary care during the 20-year period between 1995 and 2015 in the UK.

## METHODS
### Data source
Population-based primary care data from the Clinical Practice Research Datalink (CPRD) were used to identify a cohort of patients with a shoulder dislocation aged 16–70 years during 1995–2015 in the UK. The design of this study was approved by the Independent Scientific Advisory Committee (ISAC 15_260) for the Medicines & Healthcare products Regulatory Agency.

The CPRD covers 11.3 million people from 674 UK general practices, with a current representative coverage of approximately 6.9% of the UK population who are broadly representative of the UK population in terms of age, sex and ethnicity.[23] While the data are anonymised at the patient and general practice level, demographic information at the patient level on age, gender, geographic region, body mass index (BMI, kg/m$^2$), smoking status (current smoker, ex-smoker and non-smoker) and drinking status (current drinker, ex-drinker and non-drinker) was available. Data on the Index of Multiple Deprivation 2004[24] for English patients were linked with the CPRD data. Charlson Comorbidity Index was calculated using a list of predefined CPRD READ codes. CPRD population data for the whole CPRD dataset by individual year, age and gender were also obtained.

### Participants
Shoulder dislocations were identified using READ codes as defined a priori after consensus by specialist shoulder clinicians experienced in clinical practice and epidemiological research (see online supplementary table A). To ensure that only primary shoulder dislocations had been captured, patients had to have no recorded shoulder dislocations in the 2 years of clinical data (a 2-year washout period was defined using the date the general practitioner (GP) practice was classified as 'up-to-standard' and the date the patient first registered at the GP practice) preceding the identified shoulder dislocation. A predefined set of exclusion criteria were applied to the CPRD dataset (figure 1).

### Validation of the coding of shoulder dislocations in CPRD
A separate internal validation exercise of the coding of shoulder dislocations within CPRD was conducted among the 6046 patients aged 16–35 years who were diagnosed during 1 April 1997–31 March 2015 in England using the same READ codes for shoulder dislocations, as well as the same 2-year washout period. The age range used for this validation exercise was not entirely inclusive of the present study, but tested the same coding used in the older age group. The 16–35 years age range was used to inform both this study and a future commissioned study that will include linked hospital data among patients aged 16–35 years because this group currently has the highest level of priority for clinical interventions.

A GP questionnaire was designed based on a validation algorithm (see online supplementary table B) and with the assistance of GPs. A random sample of 172 patients was selected, and CPRD personnel sent the questionnaire to the patient's GP practice for a clinician to complete by comparing the records on their CPRD system with the patient's clinical records. Four written reminders were sent every 2 weeks to the GP practices. Data from the questionnaires were double-entered into a dataset by a statistician and a project manager, and queries were resolved by an academic orthopaedic shoulder surgeon.

The following validation criteria were stipulated a priori for proceeding with shoulder dislocation research studies using CPRD data:
A. A positive predictive value of accurate shoulder dislocation coding within CPRD of at least 75%.

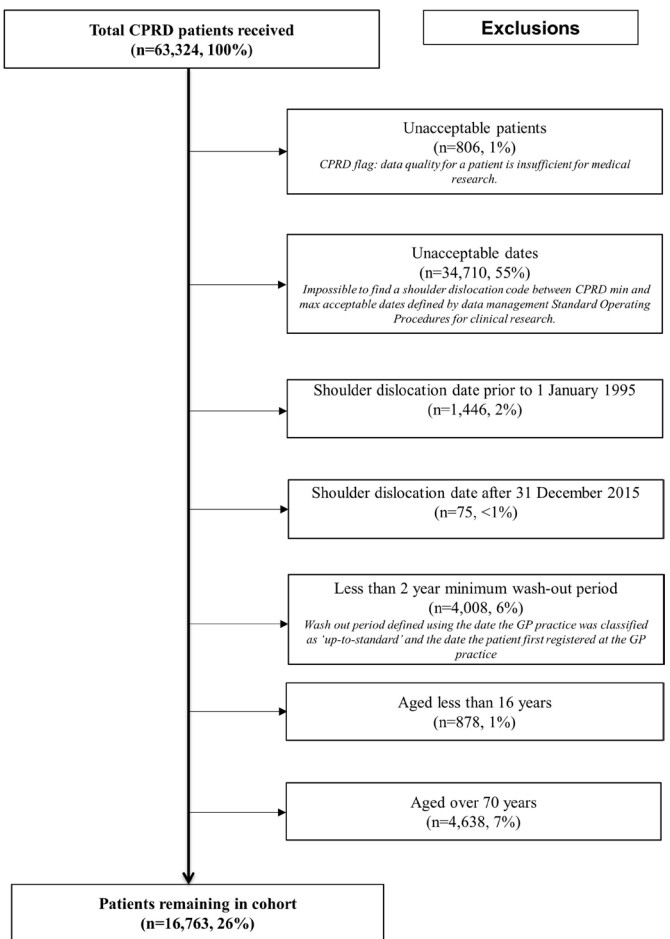

**Figure 1** Shoulder dislocation exclusion flow chart for patients aged 16–70 years during 1995–2015 within the CPRD, UK. CPRD, Clinical Practice Research Datalink; GP, general practitioner.

B. A positive predictive value of accurate 'primary' or 'first time' shoulder dislocation coding within CPRD of at least 75%.

### Statistical analysis of the data

Descriptive statistics were used to summarise the epidemiology of primary shoulder dislocations by demographic factors. Incidence rates by age and gender per 100 000 person-years and incidence rate ratios with 95% CIs and P values were calculated using STATA V.14.1.[25]

Denominators for the incidence rates were constructed using patient-level data from CPRD. Observation time per patient was calculated between 1995 and 2015 as the sum of total year time contributed by all subjects where person-years start as the latest of first registration date, practice up-to-standard date and 1 January 1995 and end as the earliest among patient transfer out date, practice last collection date, death date and 31 December 2015.

### RESULTS
### Validation of the coding of shoulder dislocations within CPRD

Of the 172 patients for whom a GP validation questionnaire was sent, we received responses for 95 (55%)

patients. For an additional two patients, GPs confirmed that the patient had transferred out of the practice, and no medical records or information on the CPRD system was available for them. The demographic characteristics of: (A) the CPRD cohort aged 16–35 years during 1 April 1997–31 March 2015 in England (n=6046), (B) 172 patients randomly selected to have a questionnaire sent to their GP for additional information and (C) the responders and (D) non-responders to the GP questionnaire are given in table 1.

Characteristics such as age, BMI and Charlson Comorbidity Index were similar across all groups. A 100% response rate was received from GP practices in the South West, but aside from this, responses reflected the regional distribution of patients in the whole cohort. Subsequently linked data on the Index of Multiple Deprivation 2004 revealed that a higher proportion of patients in categories 1 (affluent) and 4 (somewhat deprived) had been sampled than in the population of 6046 patients, but otherwise response rates were similar from all deprivation groups.

Shoulder dislocation was confirmed as having been coded correctly within CPRD for 89% (95% CI 83% to 95%) of all patients (table 2). GPs confirmed that the remaining 11% (10 patients) had been miscoded and that the patient had suffered other shoulder trauma or injuries such as capsulitis and strains or dislocations of the acromioclavicular joint. Of all patients with a shoulder dislocation, 76% (95% CI 67% to 85%) were confirmed as having had a 'primary' or first time shoulder dislocation. A further dislocation, occurring up to 2 years after the primary dislocation, was recorded within CPRD for 32% of patients who had had a confirmed shoulder dislocation. However, an additional 11% of patients had a further dislocation which was not recorded within CPRD.

### Incidence of shoulder dislocations in the UK

An initial cohort of 63 324 patients with shoulder dislocation codes was identified, and then a predefined set of exclusion criteria were applied (figure 1). Most patients were excluded (55%) because they had been diagnosed outside the study time period or because they were either younger than 16 years or older than 70 years (8%).

A cohort of 16 763 patients with first time anterior shoulder dislocations aged 16–70 years in the UK during 1995–2015 remained in the CPRD dataset and was used in further analyses. Numbers of patients identified by CPRD READ code are presented in online supplementary table A. The baseline characteristics of the cohort are given in table 3. Most (72%) shoulder dislocations occurred in men, and the median age for the whole cohort was 36 years (IQR 24–52 years). Most patients had a 'normal' (18.5–24.9 kg/m²) BMI, and 88% had no comorbidities.

The percentage of patients with primary shoulder dislocation by age and gender in the UK is given in figure 2. The peak in numbers for men is spread over those aged 17–22 years, and a peak was observed in women aged 61–70 years.

**Table 1** Demographic characteristics of patients with shoulder dislocation aged 16–35 years recorded within the CPRD dataset during 1 April 1997–31 March 2015 in England and responders and non-responders to the CPRD GP validation questionnaire

| Demographic characteristic | Whole cohort* | All GP questionnaires* | Responders* | Non-responders* |
|---|---|---|---|---|
| Cohort size | 6046 | 172 | 97 | 75 |
| Gender | | | | |
| Men | 4991 (83%) | 137 (80%) | 81 (84%) | 56 (75%) |
| Women | 1055 (17%) | 35 (20%) | 16 (16%) | 19 (25%) |
| Median age, years (IQR) | 24 (20–34) | 24 (20–29) | 24 (20–29) | 24 (19–29) |
| Median BMI, kg/m$^2$ (IQR) | 24 (22–27) | 24 (21–27) | 25 (22–28) | 23 (21–26) |
| Median Charlson Comorbidity Index (IQR) | 0 (0–0) | 0 (0–0) | 0 (0–0) | 0 (0–0) |
| Region | | | | |
| East Midlands | 263 (4%) | 0 (0%) | 0 (0%) | 0 (0%) |
| East of England | 673 (11%) | 21 (12%) | 17 (18%) | 4 (5%) |
| London | 695 (12%) | 23 (13%) | 11 (11%) | 12 (16%) |
| North East | 133 (2%) | 4 (2%) | 3 (3%) | 1 (1%) |
| North West | 951 (16%) | 29 (17%) | 13 (13%) | 16 (21%) |
| South Central | 965 (16%) | 32 (19%) | 17 (18%) | 15 (20%) |
| South East Coast | 702 (12%) | 25 (15%) | 12 (12%) | 13 (17%) |
| South West | 743 (12%) | 17 (10%) | 17 (18%) | 0 (0%) |
| West Midlands | 667 (11%) | 15 (9%) | 7 (7%) | 8 (11%) |
| Yorkshire and the Humber | 254 (4%) | 6 (3%) | 0 (0%) | 6 (8%) |
| Index of Multiple Deprivation 2004 | | | | |
| Affluent | 1279 (21%) | 53 (31%) | 28 (29%) | 26 (35%) |
| 2 | 1077 (18%) | 35 (20%) | 20 (21%) | 15 (20%) |
| 3 | 958 (16%) | 24 (14%) | 16 (16%) | 8 (11%) |
| 4 | 876 (14%) | 38 (22%) | 19 (20%) | 18 (24%) |
| Deprived | 624 (10%) | 22 (13%) | 14 (14%) | 8 (11%) |
| Missing | 1232 (20%) | 0 (0%) | 0 (0%) | 0 (0%) |

*Number and percentage (%) are presented unless otherwise stated.
BMI, body mass index; CPRD, Clinical Practice Research Datalink; GP, general practitioner.

Incidence rates and rate ratios by age and gender for patients with primary shoulder dislocation in the UK are presented in table 4. The overall incidence rate in men was 40.4 per 100 000 person-years (95% CI 40.4 to 40.4), and in women was 15.5 per 100 000 person-years (95% CI 15.5 to 15.5). The highest incidence was observed in men aged 16–20 years (80.5 per 100 000 person-years; 95% CI 80.5 to 80.6), and incidence in men decreased with increasing age (figure 3). A U-shaped pattern of incidence was observed in women; incidence was 16.4 per 100 000 person-years

**Table 2** Validation of shoulder dislocations coded within the CPRD dataset: responses to GP questionnaires (n=95*)

| | n | % |
|---|---|---|
| GP confirmation of shoulder dislocation | 85 | 89 |
| Patients who had a confirmed 'primary' shoulder dislocation | 72 | 76 |
| Patients who had a further dislocation within 2 yearsof the primary dislocation† | 27 | 32 |
| Confirmation that this was a further dislocation episode and not a review of the problem | 21 | 78 |
| Patients with further dislocations that have notbeen coded within CPRD† | 9 | 11 |

*An additional two questionnaires were received from GPs stating that the patient had transferred out of the practice and that no data were available for them. We have omitted these two patients from the denominator used for this table.
†The denominator is 85 because these patients were confirmed to have had a shoulder dislocation and could then potentially have a redislocation.
CPRD, Clinical Practice Research Datalink; GP, general practitioner.

**Table 3** Baseline characteristics of patients with primary shoulder dislocation aged 16–70 years within the Clinical Practice Research Datalink dataset during 1995–2015, UK

| Characteristic | n | % |
|---|---|---|
| Total | 16 763 | 100 |
| Gender | | |
| Men | 12 148 | 72 |
| Women | 4615 | 28 |
| Age at shoulder dislocation (years) | | |
| 16–20 | 2561 | 15 |
| 21–30 | 4266 | 25 |
| 31–40 | 3021 | 18 |
| 41–70 | 6915 | 41 |
| Body mass index (kg/m$^2$) | | |
| <18.5 | 180 | 1 |
| 18.5–24.9 | 3392 | 20 |
| 25.0–29.9 | 3020 | 18 |
| 30.0–34.9 | 1292 | 8 |
| ≥35.0 | 768 | 5 |
| Missing | 8111 | 48 |
| Smoking | | |
| Non-smoker | 6674 | 40 |
| Current smoker | 3388 | 20 |
| Ex-smoker | 2014 | 12 |
| Missing | 4687 | 28 |
| Drinking | | |
| Current drinker | 6854 | 41 |
| Non-drinker | 1113 | 7 |
| Ex-drinker | 188 | 1 |
| Missing | 8608 | 51 |
| Charlson Comorbidity Index | | |
| 0 | 14 834 | 88 |
| 1 | 950 | 6 |
| 2 | 523 | 3 |
| ≥3 | 456 | 3 |
| Region | | |
| East Midlands | 600 | 4 |
| East of England | 1444 | 9 |
| London | 1484 | 9 |
| North East | 279 | 2 |
| North West | 2071 | 12 |
| Northern Ireland | 602 | 4 |
| Scotland | 1626 | 10 |
| South Central | 2005 | 12 |
| South East Coast | 1572 | 9 |
| South West | 1462 | 9 |

Continued

**Table 3** Continued

| Characteristic | n | % |
|---|---|---|
| Wales | 1591 | 9 |
| West Midlands | 1470 | 9 |
| Yorkshire and the Humber | 557 | 3 |
| Index of Multiple Deprivation 2004 (quintile of deprivation) | | |
| Affluent | 2790 | 17 |
| 2 | 2345 | 14 |
| 3 | 2001 | 12 |
| 4 | 1793 | 11 |
| Deprived | 1309 | 8 |
| Missing | 6525 | 39 |

in those aged 16–20 years that decreased among those aged 21–50 years and then increased to 28.6 per 100 000 person-years among those aged 61–70 years (figure 3). Incidence was significantly higher in men than women in almost all age groups, with an overall incidence rate ratio of 2.60 (95% CI 2.52 to 2.69). The only exception was men and women aged 61–70 years, where no significant difference in incidence was observed (P=0.334).

## DISCUSSION
### Main findings
A large population-based cohort of 16 763 patients aged 16–70 years in the UK during 1995–2015 was identified in the CPRD dataset. A separate validation exercise demonstrated that CPRD is an acceptable dataset to identify and study patients with shoulder dislocation. Most shoulder dislocations occurred in men (72%). The overall incidence rate in men was 40.4 per 100 000 person-years, and in women was 15.5 per 100 000 person-years. The highest incidence was observed in men aged 16–20 years (80.5 per 100 000 person-years). An unexpected finding was that incidence in women increased beyond 50 years to 28.1 per 100 000 person-years among those aged 61–70 years, but this pattern was not observed in men.

### Comparison with other studies
The UK cohort was similar in age and gender distribution and incidence patterns to those observed in the Canadian, US and Oslo cohorts.[19 20 22] While incidence patterns were similar between countries, in the UK, the peak in numbers for men is spread over those aged 17–22 years, whereas there is a distinct peak in men aged 17–18 years in Canada and the USA. Possible reasons for this difference may be the high numbers of young men playing ice hockey and American Football at school aged 17–18 years and not all continuing to play at college. In a small study of the causes of shoulder dislocations in Sweden, incidence was high (8%) among ice hockey players.[10] Other explanations

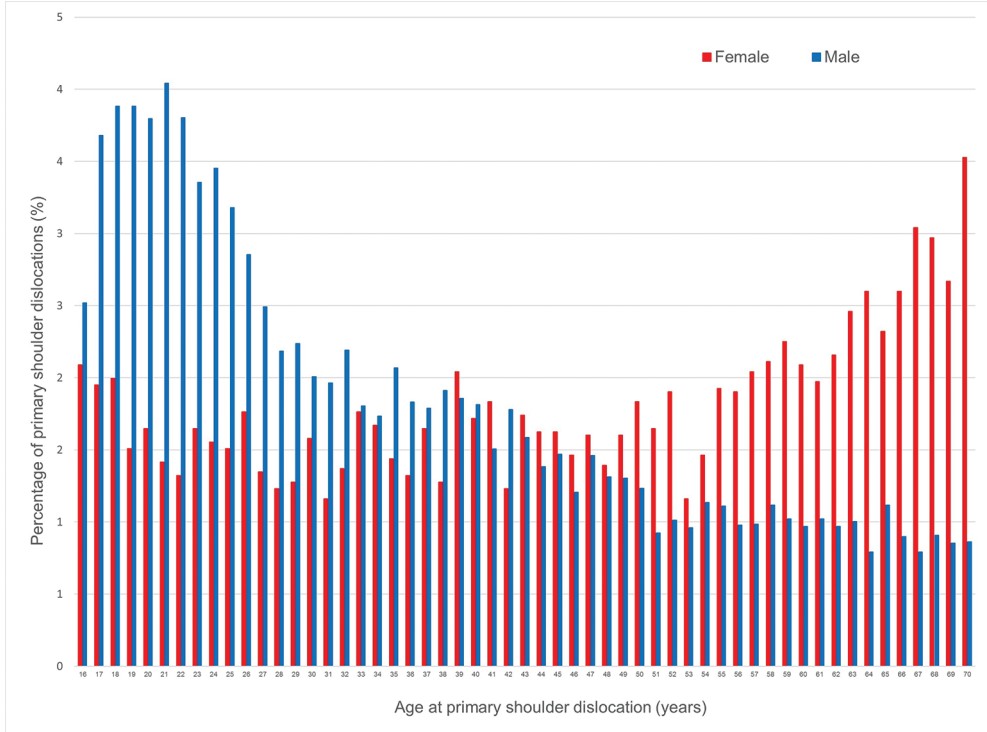

**Figure 2** Percentage of patients with primary shoulder dislocation by age (16–70 years) and gender recorded within the Clinical Practice Research Datalink during 1995–2015, UK. Blue bars represent male patients, and red bars represent female patients. 95% CIs are included in this figure, but they are so close to the main data points that they cannot be seen.

might be under-reporting of shoulder dislocations among college students or a genuine decrease due to better skeletal maturity and shoulder muscle strength and control.

Incidence rates were higher in the UK than Canada for all combinations of age groups and gender, except for men aged 16–20 years (UK men: 80.5 per 100 000 person-years vs Canadian men: 98.3 per 100 000 person-years). The higher incidence rates may be explained by UK data being based on primary care records in contrast with the Canadian data being based on hospital surgical records.[22]

A study conducted in Denmark also identified the same bimodal age distribution of incidence and also specifically noted that older people most frequently dislocated their shoulder at home by falling on their arm, whereas young people most frequently suffered a shoulder dislocation while playing sports.[2]

The increasing incidence of shoulder dislocations among women over 50 years of age is a new finding that is of both interest and concern because the reasons for it are not known. We know such injuries in the elderly are usually associated with rotator cuff tears and fractures with subsequent loss of function as well as instability. However, further work is required to examine the reasons that may underpin this increased risk of shoulder dislocations in ageing women. Possible reasons include biological differences between ageing men and women such as differences in joint proprioception, soft tissue tendon quality, protective muscle bulk or whether there is a difference in

the incidence of falls between men and women. This is of particular importance given that the population of the UK continues to change to include more elderly people. With the increasing population, priority needs to be given to increasing the safety of the elderly to reduce falls, dislocations and fractures, as advocated by the National Institute for Health and Care Experience.[26]

### Strengths and limitations

The main strength of this study is the large population-based cohort using real-world data from primary care. CPRD is representative of the UK general population by age and gender. The age-specific and gender-specific incidence of primary shoulder dislocations in the UK are similar to those observed in Canada and the USA, which supports the validity of the patterns observed. The incidence of traumatic shoulder dislocations in these other countries, however, has only been calculated using regional data or hospital data, and so this is the first time the incidence of shoulder dislocations has been studied using population-based primary care data, and the first time results for the UK have been produced.

Another strength of this study was the initial validation of shoulder coding in CPRD. Previous studies have successfully adopted the same approach when using CPRD data.[27 28] The validity of GP coding of shoulder dislocations within CPRD among a subset of patients proved very high at 89%, and of these, 76% were confirmed to be 'primary' shoulder dislocations. All of the CPRD READ codes used to

**Table 4** The number, incidence rates and incidence rate ratios of primary shoulder dislocation by age and gender within the Clinical Practice Research Datalink dataset during 1995–2015, UK

| Demographic category | n | Person-years* | Incidence rate† | 95% CI | Demographic comparison | Incidence rate ratio | 95% CI | P value |
|---|---|---|---|---|---|---|---|---|
| **Gender** | | | | | | | | |
| Male | 12 148 | 30 074 078 | 40.4 | 40.4–40.4 | Male versus female | 2.60 | 2.52–2.69 | <0.001 |
| Female | 4615 | 29 741 559 | 15.5 | 15.5–15.5 | | | | |
| **Age (years)** | | | | | | | | |
| 16–20 | 2561 | 5 245 428 | 48.8 | 48.8–48.9 | | | | |
| 21–30 | 4266 | 11 006 586 | 38.8 | 38.7–38.8 | 16–20 vs 21–30 | 1.26 | 1.20–1.32 | <0.001 |
| 31–40 | 3021 | 12 362 061 | 24.4 | 24.4–24.5 | 16–20 vs 31–40 | 2.00 | 1.90–2.11 | <0.001 |
| 41–50 | 2472 | 12 244 890 | 20.2 | 20.2–20.2 | 16–20 vs 41–50 | 2.42 | 2.29–2.56 | <0.001 |
| 51–60 | 2091 | 10 583 309 | 19.8 | 19.7–19.8 | 16–20 vs 51–60 | 2.47 | 2.33–2.62 | <0.001 |
| 61–70 | 2352 | 8 373 363 | 28.1 | 28.1–28.1 | 16–20 vs 61–70 | 1.74 | 1.64–1.84 | <0.001 |
| **Age, gender (male)** | | | | | | | | |
| 16–20 | 2137 | 2 653 062 | 80.5 | 80.5–80.6 | | | | |
| 21–30 | 3588 | 5 463 830 | 65.7 | 65.6–65.7 | 16–20 vs 21–30 | 1.23 | 1.16–1.29 | <0.001 |
| 31–40 | 2316 | 6 265 348 | 37.0 | 36.9–37.0 | 16–20 vs 31–40 | 2.18 | 2.05–2.31 | <0.001 |
| 41–50 | 1733 | 6 243 377 | 27.8 | 27.7–27.8 | 16–20 vs 41–50 | 2.90 | 2.72–3.09 | <0.001 |
| 51–60 | 1244 | 5 342 095 | 23.3 | 23.3–23.3 | 16–20 vs 51–60 | 3.46 | 3.22–3.71 | <0.001 |
| 61–70 | 1130 | 4 106 366 | 27.5 | 27.5–27.5 | 16–20 vs 61–70 | 2.93 | 2.72–3.15 | <0.001 |
| **Age, gender (female)** | | | | | | | | |
| 16–20 | 424 | 2 592 366 | 16.4 | 16.3–16.4 | | | | |
| 21–30 | 678 | 5 542 756 | 12.2 | 12.2–12.2 | 16–20 vs 21–30 | 1.34 | 1.18–1.51 | <0.001 |
| 31–40 | 705 | 6 069 714 | 11.6 | 11.6–11.6 | 16–20 vs 31–40 | 1.41 | 1.25–1.60 | <0.001 |
| 41–50 | 739 | 6 001 514 | 12.3 | 12.3–12.3 | 16–20 vs 41–50 | 1.33 | 1.18–1.50 | <0.001 |
| 51–60 | 847 | 5 241 214 | 16.2 | 16.1–16.2 | 16–20 vs 51–60 | 1.01 | 0.90–1.14 | 0.840 |
| 61–70 | 1222 | 4 266 996 | 28.6 | 28.6–28.7 | 16–20 vs 61–70 | 0.57 | 0.51–0.64 | <0.001 |
| **Age, gender (male vs female)** | | | | | | | | |
| | | | | | 16–20 vs 16–20 | 4.92 | 4.44–5.47 | <0.001 |
| | | | | | 21–30 vs 21–30 | 5.37 | 4.95–5.83 | <0.001 |
| | | | | | 31–40 vs 31–40 | 3.20 | 2.94–3.48 | <0.001 |
| | | | | | 41–50 vs 41–50 | 2.25 | 2.07–2.46 | <0.001 |
| | | | | | 51–60 vs 51–60 | 1.44 | 1.32–1.57 | <0.001 |
| | | | | | 61–70 vs 61–70 | 0.96 | 0.89–1.04 | 0.334 |

*Person-years used as the denominator for incidence rates, as obtained for patients aged 16–70 years during 1995–2015 in the CPRD dataset.
†The incidence rate per 100 000 person-years.
CPRD, Clinical Practice Research Datalink.

identify patients with shoulder dislocation were useful for identifying patients who had a primary shoulder dislocation, including the three codes that are specific for redislocations (N083A00, N083100 and N083C00).

While it is possible that someone suffering their first shoulder dislocation may not seek medical care, the rate of redislocation is so high that it seems very unlikely that such patients would not visit their GP with this diagnosis within 2 years and the medical history section of the clinical record updated. In epidemiological studies, a threshold of 70% is often used for correctly identifying a diagnosis. Thus, 76% specificity of identifying primary shoulder dislocations within CPRD is a very satisfactory level for conducting epidemiological analyses.

A possible limitation of the validation exercise is that it only included patients aged 16–35 years; however, we are

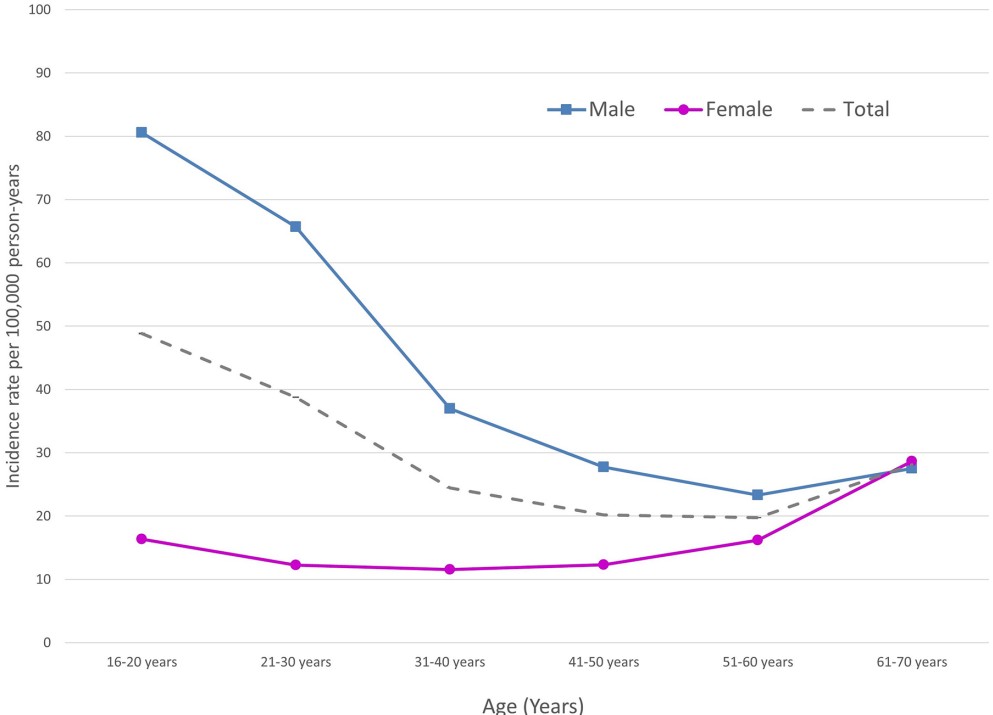

**Figure 3** Shoulder dislocation incidence rates by age and gender during 1995–2015 in the UK, using data from the Clinical Practice Research Datalink.

unaware of any reasons why coding of shoulder dislocations would be different for patients aged up to 70 years. While not all GP practices responded to the questionnaire, those that did and did not respond to the questionnaire survey were similar by deprivation, geography and other demographic characteristics.

## CONCLUSIONS

In the UK, most primary shoulder dislocations occurred in young men. An unexpected finding was that incidence increased beyond the age of 50 years in women, but not in men. The reasons for this are unknown. Priority and attention should be directed to increasing preventative measures for young people playing contact sports and to research on the possible causes of the increase in incidence for women after the age of 50 years.

This data will now facilitate future commissioned research to determine the predictors of recurrent dislocations and to compare recurrence of shoulder dislocation in those treated with surgery compared with those not having surgery.

**Acknowledgements** The authors would like to express their gratitude for the work of Margaret Smith and Clare Bankhead at the Primary Care Unit of the University of Oxford for facilitating access to Clinical Practice Research Datalink data on shoulder dislocations. The authors would also like to express their gratitude for the work of Robin May and Emma Boyle at the Clinical Practice Research Datalink for facilitating the internal validation study using GP questionnaires.

**Contributors** AS analysed the data and drafted the paper. AJ assisted with the analysis and commented on the paper. AD assisted with managing and cleaning the data and commented on the paper. KE assisted with the literature review, data access and internal validation of the study and commented on the paper. NKA, DP-

A, AJC, TAH, RAP-V, SH, SEL, AR and GC commented on the paper. JR designed the study, oversaw the analysis and revised the paper.

**Funding** This project was funded by the National Institute of Health and Research (NIHR) Health Technology Assessment (HTA) Programme (project number: 14/160). The research was supported by the National Institute for Health Research (NIHR) Oxford Biomedical Research Centre (BRC).

**Disclaimer** The views and opinions expressed therein are those of the authors and do not necessarily reflect those of the NHS, the NIHR or the Department of Health. All researchers are independent from the study funders. This study is based in part on data from the Clinical Practice Research Datalink obtained under licence from the UK Medicines and Healthcare products Regulatory Agency. The funding source had no role in the design and conduct of the study, in the collection, analysis and interpretation of the data, or in the preparation, review or approval of the manuscript.

**Competing interests** AJ has received a grant from NIHR HS & DR during the conduct of the study, and has received consultancy, lecture fees and honoraria from Servier, UK Renal Registry, Oxford Craniofacial Unit, IDIAP Jordi Gol, Freshfields Bruckhaus Deringer, has held advisory board positions (which involved receipt of fees) from Anthera Pharmaceuticals, Inc., and received research sponsorship from ROCHE. NKA has received honoraria, held advisory board positions (which involved receipt of fees) and received consortium research grants from Merck (honorarium); Roche, Novartis and Bioiberica (grants); Smith & Nephew, Nicox, Flexion Bioventus and Freshfields (personal fees) outside the submitted work. DP-A has received research grants from Amgen and Laboratoires Servier. TH is a GP in London, and is a GP advisor for, but not employed by, the Clinical Practice Research Datalink (CPRD). AR has received personal fees from DePuy Ltd., Aviva Health, AXA PPP Healthcare, AXA PPP International, BUPA, Cigna, SecureHealth, Simplyhealth, Vitality Health (Pru Health) and WPA; a grant from JRI Ltd.; and has a patent for a shoulder replacement prosthesis.

**Ethics approval** The Independent Scientific Advisory Committee (ISAC)

**Provenance and peer review** Not commissioned; externally peer reviewed.

**Data sharing statement** Data are not available for sharing.

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
