## [Reviewer comments · BMJ Open]

ARTICLE DETAILS

TITLE (PROVISIONAL)	Incidence of shoulder dislocations in the United Kingdom, 1995-2015: a population-based cohort study
AUTHORS	Shah, Anjali; Judge, Andrew; Delmestri, Antonella; Edwards, Katherine; Arden, Nigel; Prieto-Alhambra, Daniel; Holt, Tim; Pinedo-Villanueva, Rafael; Hopewell, Sally; Lamb, Sarah; Rangan, Amar; Carr, Andrew; Collins, Gary; Rees, Jonathan

VERSION 1 – REVIEW

REVIEWER	Kim Barber Foss Cincinnati Childrens Hospital USA
REVIEW RETURNED	08-Jun-2017

GENERAL COMMENTS	I applaud such a large undertaking. I would request that more work should be placed on details in the methods section and better and more concise flow of the manuscript. One concern I have is that you state several times that women's incidence increases with age, however your data showed a horseshoe shape, so "increasing with age" really is not accurate when there is a decrease in the middle ages.
--

REVIEWER	Matthew J. Kraeutler, MD Seton Hall-Hackensack Meridian School of Medicine, South Orange, NJ, USA
REVIEW RETURNED	29-Jun-2017

GENERAL COMMENTS	Abstract: Define GP, NIHR Introduction: Other studies have shown that anterior dislocations comprise about 80% of all shoulder dislocations (Owens et al, 2007). Please include a range and more references. Methods: Page 10, line 23: Can you explain why you validated only patients aged 16-35 years? Results: Pg 14, line 34: Was this confirmed for 76% of all primary dislocations? Or for 76% of ALL dislocations?
---

	Did you do any statistical analysis based on the demographics shown in Table 3 (smoking, drinking, BMI, etc)? Discussion: Page 21, last paragraph: Please do some further research to look into these possible reasons for the differences between males and females? Is there a known difference in risk of falls or joint proprioception?
--	---

VERSION 1 – AUTHOR RESPONSE

Reviewer 1:

Comments and Responses:

1. I applaud such a large undertaking. I would request that more work should be placed on details in the methods section and better and more concise flow of the manuscript.

Author response: We thank the reviewer for this comment. We have been through the manuscript in detail (also incorporating Reviewer 2's suggestions) and we have added detail to the Methods and made changes to the whole manuscript with brevity in mind.

2. One concern I have is that you state several times that women's incidence increases with age, however your data showed a horseshoe shape, so "increasing with age" really is not accurate when there is a decrease in the middle ages.

Author response: Thank you for drawing our attention to the lack of specificity on this point. We have amended five sentences to read:

Abstract Conclusion

"While most primary dislocations occurred in young men, an unexpected finding was that incidence increased in women aged over 50 years, but not in men."

Strengths and limitation of this study second bullet point, immediately following the Abstract

- "A new finding is that shoulder dislocations in women over the age of 50 years increased to a peak of 28.6 per 100,000 person-years in those aged 61-70 years"

Main findings, first paragraph

"An unexpected finding was that incidence in women increased beyond 50 years to 28.1 per 100,000 person-years among those aged 61-70 years, but this pattern was not observed in men."

Comparison with other studies, fourth paragraph

"The increasing incidence of shoulder dislocations among women over 50 years of age is a new finding that is of both of interest and concern because the reasons for it are not known."

Conclusions, first paragraph

"In the UK, most primary shoulder dislocations occurred in young men. An unexpected finding was that incidence increased beyond the age of 50 years in women, but not in men."

Reviewer 2:

Comments and Responses:

3. Abstract: Define GP, NIHR

Author response: The abstract has been amended as requested to read 'general practitioner (GP)' and 'National Institute for Health Research (NIHR)'.

4. Introduction: Other studies have shown that anterior dislocations comprise about 80% of all shoulder dislocations (Owens et al, 2007). Please include a range and more references.

Author response: Thank you for this constructive comment. We have amended the text to include a range and referred to 5 additional journal articles. The text now reads:

"Shoulder joint dislocations are the most common joint dislocations seen in hospital Accident and Emergency departments and trauma clinics 1, 2 with 80-97% of traumatic glenohumeral dislocations being anterior. 1-6"

5. Methods: Page 10, line 23: Can you explain why you validated only patients aged 16-35 years?

Author response: The primary reason for conducting the validation in this age group is because the peak of incidence occurs in this age group and this group currently have the highest level of priority for clinical interventions.

A future commissioned study on the outcome for traumatic anterior shoulder dislocations who have surgery compared with those who have no surgery is specifically going to involve 16-35 years-olds who were diagnosed during 1 April 1997 – 31 March 2015. Despite the difference in age ranges in this study and the future study, we believe the results of the validation exercise are relevant. We have no reason to expect any differences in shoulder dislocation coding by GP practices for patients over the age of 35 years.

We have amended the text in the Methods to read:

"The age range used for this validation exercise was not entirely inclusive of the present study, but tested the same coding used in the older age group. The 16-35 years age range was used to inform both this study and a future commissioned study that will include linked hospital data among patients aged 16-35 because this group currently have the highest level of priority for clinical interventions."

6. Results: Pg 14, line 34: Was this confirmed for 76% of all primary dislocations? Or for 76% of ALL dislocations?

Author response: 76% of all shoulder dislocations were confirmed as a 'primary'.

We have amended this sentence to read: "Of all patients with a shoulder dislocation, 76% (95% CI: 67%-85%) were confirmed as having had a 'primary' or first time shoulder dislocation."

7. Results: Did you do any statistical analysis based on the demographics shown in Table 3 (smoking, drinking, BMI, etc)?

Author response: The aim of this study was purely a descriptive assessment of incidence trends by age and gender.

The focus of future research will include additional variables. In a subsequent study we will evaluate these factors as potential confounders for any association between surgery and shoulder re-dislocation.

Also, there is a high level of missing data for smoking, drinking and BMI, which would bring into question the validity of incidence estimates by their sub-categories, and we do not have population denominators for these variables.

8. Discussion: Page 21, last paragraph: Please do some further research to look into these possible reasons for the differences between males and females? Is there a known difference in risk of falls or joint proprioception?

Author response: We agree that a high priority should be given to investigating the reasons for this gender difference. The data we currently hold do not include information on falls, mental health conditions or neural problems that might influence joint proprioception. As such all we can do at present is highlight this finding and suggest further research is required.

We have amended a sentence in the Conclusion to read:

“Priority and attention should be directed to increasing preventative measures for young people playing contact sports, and to research on the possible causes of the increase in incidence for women after the age of 50 years.”

VERSION 2 – REVIEW

REVIEWER	Kim barber Foss CCHMC, United States
REVIEW RETURNED	22-Aug-2017

GENERAL COMMENTS	Please review for grammar. also Line 32 - multiple "had".
---

REVIEWER	Ruth Pickering University of Southampton UK
REVIEW RETURNED	27-Sep-2017

GENERAL COMMENTS	I thought this was an interesting paper. I have the following points to raise. 1 Statistical analysis of data, page 11, line 29/30. They should include a sentence about the calculation of person years, including all ways in which a patient's person years during the study period 1995-2015 could start, and how it could end. 2 Page 11, the sentence starting on line 45, sounds like methods not results and repeats information from the sentence starting on page 10, line 45/46. In neither place is it explained what the total of 6042 relates to.
--

	3 Page 16, lines 5-10, the relationship between increasing affluence and risk of shoulder dislocation. There are no statistics reported supporting this statement. 4 Page 17 line 53/4 and page 18 line 3. The reference should be to Figure 3 not Figure 2. In Table 4 incidence rates etc, are presented for age groups, 16-20, 21-30, 31-40, an 41-70, but in Figure 3 the final age group is split 41-70, 51-60 and 61-70. Table 4 wouldn't be that much larger if the same more detailed age bands were presented for consistency. Unless there is some reason to present alternative groupings of the older ages - but if so it is not mentioned anywhere. 5 Table 4, demographic comparison and incidence rate ratio columns. Ordinarily rate ratios are presented compared to a single reference group (perhaps the largest group) in the denominator. Here rate ratios are presented with the 16-20 group in most of the numerators and different comparison age groups in the denominators. Also the row of the table on which two rates are contrasted in the ratio in most case doesn't relate to either age group being compared. Most confusing. These incidence rate ratios are not referred to in the results, and there is no explanation of the eccentric presentation in the statistical methods section on page 11. 6 Discussion, page 20, line 38/9. The statement that the peak numbers in men in the UK being between 17-22. This is from Fig 2. But from Table 2 the peak rate is in the category 16-20, and this seems similar to the peak age range in Canada of 17-18. So I felt that the UK and Canadian findings were similar not dissimilar. 7 Page 22, lines 36-43. The comment that all the READ codes were useful in identifying patients. There are no results supporting this in the main paper. It would be interesting to see this verified in the Results section.
--	---

VERSION 2 – AUTHOR RESPONSE

Reviewer 1

Comment: Please review for grammar. also Line 32 - multiple "had".

Response: The sentence on page 6 of 68 line 32 currently reads:

This study while small is well cited in the literature, and observed that 1.7% of the population aged 18-70 years had had a shoulder dislocation.

In UK English the use of 'had had' in this sentence is grammatically correct. However, this may confuse an international audience, and thus we have changed it to read:

This study while small is well cited in the literature, and observed that 1.7% of the population aged 18-70 years previously had a shoulder dislocation.

We would be happy to accept any further grammatical suggestions that the editors of BMJ Open make to suit the audience of the journal.

Reviewer 2:

Comment and Responses:

1 Statistical analysis of data, page 11, line 29/30. They should include a sentence about the calculation of person years, including all ways in which a patient's person years during the study period 1995-2015 could start, and how it could end.

Response: We thank the reviewer for this comment and we have added the following paragraph to the Statistical analysis of the data section:

"Denominators for the incidence rates were constructed using patient-level data from CPRD. Observation time per patient was calculated between 1995 and 2015 as the sum of total year time contributed by all subjects where person years start as the latest of first registration date, practice up-to-standard date and 1 January 1995, and end as the earliest among patient transfer out date, practice last collection date, death date and 31 December 2015."

2 Page 11, the sentence starting on line 45, sounds like methods not results and repeats information from the sentence starting on page 10, line 45/46. In neither place is it explained what the total of 6042 relates to.

Response: We appreciate the request for clarity on this issue. The section in the Methods titled "Validation of the coding of shoulder dislocations in CPRD" now has a first sentence that reads:

"A separate internal validation exercise of the coding of shoulder dislocations within CPRD was conducted among the 6,046 patients aged 16-35 years who were diagnosed during 1 April 1997-31 March 2015 in England using the same READ codes for shoulder dislocations, as well as the same two-year wash-out period."

The first sentence of the Results section has been deleted as it was a repetition of methodology, and the new first sentence has been amended to now read:

"Of the 172 patients for whom a GP validation questionnaire was sent, we received responses for 95 (55%) patients."

3 Page 16, lines 5-10, the relationship between increasing affluence and risk of shoulder dislocation. There are no statistics reported supporting this statement.

Response: This sentence currently states:

A pattern of increasing affluence, as measured by the Index of Multiple Deprivation 2004, and increasing numbers of shoulder dislocations were observed.

We were merely describing the data and not looking for a relationship with risk of shoulder dislocation, because all the patients in our study have shoulder dislocations. We cannot compare those with and without shoulder dislocations, and thus no statistics can be reported. To avoid confusion, we have deleted this sentence.

4 Page 17 line 53/4 and page 18 line 3. The reference should be to Figure 3 not Figure 2. In Table 4 incidence rates etc, are presented for age groups, 16-20, 21-30, 31-40, and 41-70, but in Figure 3 the final age group is split 41-70, 51-60 and 61-70. Table 4 wouldn't be that much larger if the same more detailed age bands were presented for consistency. Unless there is some reason to present alternative groupings of the older ages - but if so it is not mentioned anywhere.

Response: We have corrected the reference to Figure 3 as requested.

Thank you for pointing out this inconsistency in the age groups reported between Table 4 and Figure 3. The reason for the choice of age groupings in Figure 3 and Table 4, was to enable comparison with published Canadian and US data in the following references:

Leroux T, Wasserstein D, Veillette C, et al. Epidemiology of primary anterior shoulder dislocation requiring closed reduction in Ontario, Canada. *Am J Sports Med.* 2014;42(2):442-50.

Zacchilli MA, Owens BD. Epidemiology of shoulder dislocations presenting to emergency departments in the United States. *J Bone Joint Surg Am.* 2010;92(3):542-9

Response: However, we agree that consistency in our results is more important than an international comparison, and we have amended Table 4 to have the same age groupings as Figure 3.

5 Table 4, demographic comparison and incidence rate ratio columns. Ordinarily rate ratios are presented compared to a single reference group (perhaps the largest group) in the denominator. Here rate ratios are presented with the 16-20 group in most of the numerators and different comparison age groups in the denominators. Also the row of the table on which two rates are contrasted in the ratio in most case doesn't relate to either age group being compared. Most confusing. These incidence rate ratios are not referred to in the results, and there is no explanation of the eccentric presentation in the statistical methods section on page 11.

Response: As explained in the point above, we analysed the data in such a way to facilitate direct comparison with the Canadian data. However, we agree that the choice of rate ratio comparisons is unusual, and we have amended them as suggested by the reviewer. We have kept the reference group as the 16-20 year olds, so that some international comparisons can be made.

We have amended and added a sentence to the end of the Results that reads:

"Incidence was significantly higher in men than women in almost all age groups, with an overall incidence rate ratio of 2.60 (95%CI: 2.52-2.69). The only exception was men and women aged 61-70 years, where no significant difference in incidence was observed (P=0.334)."

6 Discussion, page 20, line 38/9. The statement that the peak numbers in men in the UK being between 17-22. This is from Fig 2. But from Table 2 the peak rate is in the category 16-20, and this seems similar to the peak age range in Canada of 17-18. So I felt that the UK and Canadian findings were similar not dissimilar.

Response: We agree with the reviewer that the incidence figures have greater importance than numbers, and the first sentence of the second paragraph of the Discussion does state:

"Comparison with other studies

The UK cohort was similar in age and gender distribution and incidence patterns to those observed in the Canadian, US and Oslo cohorts. 19, 20, 22"

However, we do believe it is important to draw attention to the difference in peak in numbers between the Figures in all the papers (Figure 2 in this paper, Figure 1 of the Leroux paper and Zachilli papers cited above) because of potential differences in underlying cause that is important for prevention and important for clinicians making choices about treatment.

Response: We have amended the subsequent sentence to make this explicit:

Whilst incidence patterns were similar between countries, in the UK the peak in numbers for men is spread over those aged 17-22 years, whereas there is a distinct peak in men aged 17-18 years in Canada and the US.

7 Page 22, lines 36-43. The comment that all the READ codes were useful in identifying patients. There are no results supporting this in the main paper. It would be interesting to see this verified in the Results section.

Response: We have added a column of numbers of patients by CPRD READ code used to identify them to Supplementary Table A. The following sentence has been added to the Results section entitled Incidence of shoulder dislocations in the UK:

"Numbers of patients identified by CPRD READ code are presented in Supplementary Table A."